# Age-Related Differences in How Fear, Disgust, and Sadness Influence Strategic Aspects of Arithmetic Performance

**DOI:** 10.3390/bs15121695

**Published:** 2025-12-06

**Authors:** Camille Lallement, Patrick Lemaire

**Affiliations:** Centre National de la Recherche Scientifique (CNRS), Unité Mixte de Recherche (UMR) 7077, Centre de Recherche en Psychologie et Neurosciences (CRPN), Aix-Marseille Université, 13003 Marseille, France; camille.lallement@univ-amu.fr

**Keywords:** emotion and cognition, arithmetic, problem-solving, strategies, aging

## Abstract

How different negative emotions influence cognitive processes in general, and arithmetic in particular, remains poorly understood, and even less is known about how these effects change with aging in adulthood. The present study investigated whether disgust, fear, and sadness exert distinct effects on strategy selection and execution in arithmetic, and whether these effects vary across the adult lifespan. Young and older participants were asked to choose between two strategies (Experiment 1) and to execute instructed strategies (Experiment 2) to estimate the products of two-digit multiplication problems. Interestingly, how fear, disgust, and sadness influence strategy selection and strategy execution differed in young and older adults. Discrete negative emotions differentially influenced strategic aspects of arithmetic performance in young adults, whereas none modulated strategy selection or execution in older adults. These findings have important implications for furthering our understanding of emotion–cognition interactions as well as age-related changes in these interactions.

## 1. Introduction

How do specific negative emotions influence cognition in general and arithmetic in particular, and do these effects change with aging? Negative emotions sometimes enhance and sometimes impair attention, memory, reasoning, and decision making (see [7]; [37]; [56], for reviews). Moreover, older adults often show reduced influence of negative stimuli, a well-documented positivity effect (see [3]; [5]; [44]; [54]; [60], for reviews). However, very few studies have examined the evolution of emotion-cognition links during aging in the specific domain of arithmetic, and most of them have treated emotion dimensionally, contrasting negative, positive, and neutral emotions (e.g., [33], [34]; [38]; [45], [46]). As a consequence, we ignore whether and how emotions such as fear, sadness, or disgust affect arithmetic performance, and whether young and older adults differ in these effects. The present work addressed these issues by combining a strategy perspective with a typological approach of emotions. Before outlining the logic of the present study, we first briefly review previous findings on emotion-cognition-aging relations. Then, we review previous findings on age-related differences in the effects of emotions on arithmetic performance and the proposed mechanisms responsible for these effects.

### 1.1. Age-Related Differences in Effects of Negative Emotions on Cognition

Research has shown that negative emotions influence performance across a wide variety of cognitive domains, such as attention, memory, reasoning, and decision-making (see [7]; [37]; [56], for reviews). In all these domains, negative emotions sometimes enhance and sometimes impair performance. Beneficial and deleterious effects of negative emotions on cognition are often explained through general processing mechanisms, like attentional capture ([48]; [52]; [65]). Negative emotions automatically grab attention, thereby enhancing performance when emotions align with task goals or individuals’ past experience, but impairing it when they are irrelevant. Such attentional capture may disrupt cognitive control, bias processing priorities, and consume attentional and working-memory resources.

The effects of emotions have been shown to evolve with aging. Particularly, so-called age-related positivity effects have been robustly highlighted in several general cognitive domains (see [3]; [5]; [44]; [54]; [60], for reviews), reflecting shifts in attention toward positive and away from negative stimuli. Unlike young adults, whose attention is typically drawn to negative stimuli, older adults preferentially process emotionally positive information and down-regulate negative emotions.

Although age-related differences in the effects of emotions on cognitive performance have been well documented in general cognitive domains such as attention and memory, some gaps remain. Particularly, the influence of emotions on young and older adults’ performance has been far less examined in more specific domains such as numerical cognition. Furthermore, while attention and memory are known to show significant age-related cognitive decline, arithmetic appears to be relatively unaffected (see [11]; [63], [64], for reviews). Observing age-related differences in the effects of emotions in a domain that is relatively age-invariant would suggest that these differences are indeed due to changes in attentional biases toward (or away from) emotions or in emotional regulation capacities, rather than to cognitive decline.

### 1.2. Age-Related Differences in Effects of Emotions on Arithmetic Performance

A few studies have shown that emotions also influence performance in arithmetic ([12]; [15]; [16]; [19], [20]; [28], [29]; [33], [34]; [35]; [38]; [42]; [45], [46]; [61]; [67], [68], [69]). Most of these studies have highlighted deleterious effects of irrelevant negative emotions on arithmetic performance, and authors attributed these deleterious effects to the same attentional mechanisms commonly invoked to explain effects of emotions in general cognitive domains. Irrelevant negative emotions would capture participants’ attentional resources, thereby impairing their performance on the target arithmetic task.

Very few studies have looked at how the effects of emotions in arithmetic change with aging ([33], [34]; [38]; [45], [46]). For instance, [33] ([33]) asked young and older participants to verify simple one-digit addition problems (e.g., 8 + 4 = 13, true or false?) or to estimate the product of complex two-digit multiplication problems (e.g., which is nearer to 42 × 84; 3200 or 4500?). Each problem appeared superimposed on emotionally neutral or negative pictures. Across tasks, participants were slower in the negative condition than in the neutral condition, especially on the more difficult problems. Crucially, these deleterious effects of emotions were smaller in older adults than in young adults, suggesting reduced attentional capture by negative pictures and/or more effective emotion regulation with aging.

Explanations based solely on attentional capture seemed insufficient to account for the effects of negative emotions on arithmetic performance and age-related differences in these effects. Indeed, we hypothesized that emotions, by diverting attention, would disrupt how participants perform arithmetic tasks. For example, emotions could lead participants to use fewer effective strategies or to execute these strategies less efficiently. Similarly, age-related differences in the effects of emotions on arithmetic performance could stem from the fact that emotions would have different effects on the strategies used by young adults and those used by older adults to perform arithmetic tasks.

### 1.3. Emotions and Arithmetic: A Strategy Perspective

The conceptual framework developed by [41] ([41]) offers a valuable approach for investigating mechanisms by which emotions differently influence young and older adults’ arithmetic performance. This strategy perspective is based on previous findings that people use several strategies (i.e., “a procedure or set of procedures for achieving a higher-level goal or task”; [40], p. 365) to solve a given cognitive task ([62]). Lemaire and Siegler’s framework focuses on four key dimensions of strategic behavior, repertoire (i.e., the number and types of strategies available to individuals), selection (i.e., the strategies selected for a given problem), distribution (i.e., the relative frequency with which different strategies are used), and execution (i.e., the speed and/or accuracy with which strategies are executed).

A growing number of studies in arithmetic have adopted a strategy perspective (e.g., [19]; [35]; [38]). Recently, [38] ([38]) investigated age-related differences in the effects of emotions on the strategy use in arithmetic problem-solving tasks. Young and older adults were asked to (a) solve two-digit addition problems (e.g., 32 + 56) while reporting the strategies they used on each problem, (b) estimate the products of two-digit multiplication problems (e.g., 38 × 64) by choosing between rounding-down and rounding-up strategies, and (c) execute instructed rounding-down or rounding-up strategies to estimate the products of two-digit multiplication problems (e.g., 38 × 64). Problems were presented superimposed on emotionally negative or neutral pictures. Negative emotions led (a) young adults, but not older adults, to use fewer strategies and to change how often they used available strategies, (b) both age groups to select the better strategy less often, and (c) both age groups to execute strategies more poorly, though to a lesser extent in older adults. These findings highlighted that negative emotions influenced young and older adults’ strategy selection to the same extent, and that young adults’ strategy repertoire, distribution, and execution were more influenced by emotions than those of older adults. They follow general patterns of previous findings regarding age-related changes in the effects of emotions on cognition, particularly smaller deleterious effects of negative emotions.

Previous research on emotions and arithmetic has mainly relied on a dimensional approach, which describes emotions along continuous dimensions such as valence and arousal. This approach is useful for addressing broad questions, but is less suited to examining whether distinct emotions have different effects. To answer such more specific questions, a typological approach, which considers emotions as discrete categories, is needed. Recently, [66] ([66]) asked young and older adults to verify addition and multiplication problems that were presented superimposed on emotionally neutral or negative pictures eliciting anger, disgust, or sadness. They found that young adults were more impaired by sadness than older adults while solving addition problems. Also, older adults, but not young adults, solved multiplication problems more slowly following disgust and sad pictures than following neutral pictures. Finally, anger did not affect either group’s performance. This study provides evidence that different negative emotions can have different effects on arithmetic performance and that these effects can be modulated by age. However, none of these previous studies that suggest that different negative emotions do not have the same effects on arithmetic performance adopted a strategy perspective, which is adopted here.

### 1.4. Overview of the Present Study

The main goals of the present study were to determine (a) how arithmetic performance is influenced by different negative emotions and (b) whether this influence varies with aging. To address these issues, we conducted two experiments combining a strategy perspective with a typological approach of emotions. Young and older adults were asked to choose between rounding-down and rounding-up strategies in Experiment 1 or execute the instructed rounding strategies in Experiment 2 to estimate the products of complex two-digit multiplication problems (e.g., 34 × 67). In both experiments, problems were displayed superimposed on emotionally neutral or negative pictures that induced disgust, fear, or sadness. We tested two sets of original hypotheses.

First, we hypothesized that fear, sadness, and disgust would differentially affect strategic aspects of arithmetic performance. According to Pekrun’s Cognitive Value Theory (CVT; [49], [50]; [51]), negative emotions differ in their influence depending on whether they are deactivating or activating. Deactivating emotions, such as sadness, reduces motivation to engage in cognitive tasks, whereas activating emotions, such as fear or disgust, increases vigilance and may promote engagement. Based on this framework, we predicted that sadness would impair performance more strongly than fear or disgust, leading to slower responses and/or more errors. In contrast, fear and disgust were expected to facilitate re-engagement in the arithmetic task, resulting in faster and/or more accurate performance.

Our second hypothesis concerned age-related differences in the effects of negative emotions on strategic aspects of arithmetic performance. Given that age-related positivity effects have been robustly observed in several general cognitive domains (see [3]; [5]; [44]; [54]; [60], for reviews), and more recently in arithmetic ([33]; [38]), we predicted that the effects of negative emotions on strategic aspects of arithmetic performance will be smaller in older adults than in young adults, although age differences may be modulated by the type of negative emotions.

## 2. Experiment 1. Age-Related Differences in Effects of Emotions on Strategy Selection

### 2.1. Method

#### 2.1.1. Participants

We tested 66 participants (39 young adults; 27 older adults) and 93 participants (47 young adults; 46 older adults) in Experiment 2. In both Experiments 1 and 2, young adults were undergraduate students at Aix-Marseille University, and older adults were volunteers from distinct French metropolitan areas (see Table 1 for participants’ characteristics). The target sample size was determined using an a priori power analysis (G*Power 3.1.9.4; [13]). A recent study on the effects of negative emotions on strategy selection and execution in computational estimation tasks in young and older adults found that *η*^2^*p* ranged from 0.550 to 0.690 ([38]). Using a *η*^2^*p* = 0.550, our study design of one between-participant factor (age) and two within-participant factors (emotion and strategy) could achieve 96% power with 20 participants (10 per group). To exceed this criterion and achieve power exceeding 96%, we recruited 66 participants in Experiment 1 and 93 participants in Experiment 2. In both experiments, participants provided written informed consent. These experiments received approval from the French National Ethics Committee (Ref.: SI CNRIPH 20.04.02.47414).

#### 2.1.2. Stimuli

Arithmetic problems. The stimuli were 48 multiplication problems presented in a standard form (i.e., *a* × *b*), with the operands *a* and *b* being two-digit numbers (e.g., 24 × 68, see Table 2 for the list of multiplication problems). Given previous findings in arithmetic (see [27]; [21]; [31], for overviews), we controlled the following factors: (a) no operands had 0 or 5 as unit digits; (b) digits were not repeated in the same decade or unit positions across operands (e.g., 43 × 49); (c) no digits were repeated within operands (e.g., 44 × 58); (d) no tie problems (e.g., 32 × 32) were used; and (e) operands were between 21 and 89.

Using the rounding-down strategy (i.e., RD; rounding both operands down to their nearest decades) yielded the better (i.e., closest to exact products) estimate on half the problems (e.g., doing 60 × 70 to estimate 61 × 76), while using the rounding-up strategy (i.e., RU; rounding both operands up to their nearest decades) was the better strategy (e.g., doing 40 × 60 to estimate 34 × 59) on the other problems. The unit digit was smaller than 5 in the first operand and larger than 5 in the second operand (e.g., 43 × 79) for half the problems, and the reverse for the other half (e.g., 46 × 83). Moreover, each problem was presented twice, once with an emotionally neutral picture and once with an emotionally negative picture. The problems were presented in two blocks of 48 balanced trials. Problems displayed with a neutral picture in the first block were displayed with a negative picture in the second block, while problems initially paired with a negative picture were subsequently presented with a neutral picture. Thus, within each block, half of the problems were presented with a neutral image (e.g., nature scene) and the other half with a negative image, which could induce fear (e.g., assault scene), disgust (e.g., clogged toilet), or sadness (e.g., abandoned animal). Finally, the problems of different categories (i.e., RD/RU problems × Neutral/Negative conditions) were matched on (a) the side of the larger operand (i.e., half the problems in each category had their largest operand on the left position and half on the right position), (b) the size of the correct products, and (c) the mean percentage deviations between correct products and estimates (calculated with the following formula: ([(estimate − correct product)/correct product] × 100).

Pictures. Ninety-six pictures were selected from the International Affective Pictures System (IAPS; [36]; see Table 3 for the list of pictures). Half the pictures were emotionally neutral (*mean* valence = 4.96; *SD* = 0.21, *mean* arousal = 2.62; *SD* = 0.32), and half were emotionally negative (*mean* valence = 2.64; *SD* = 0.50, *mean* arousal = 5.94; *SD* = 0.56). Among the 48 emotionally negative pictures, 16 pictures induced disgust (*mean* valence = 2.58; *SD* = 0.54, *mean* arousal = 6.03; *SD* = 0.39), 16 induced fear (*mean* valence = 2.81; *SD* = 0.61, *mean* arousal = 6.07; *SD* = 0.59), and 16 induced sadness (*mean* valence = 2.54; *SD* = 0.29, *mean* arousal = 5.72; *SD* = 0.63).[note 1] Therefore, arousal and valence of neutral and negative pictures were matched across the RD and the RU problems.

#### 2.1.3. Procedure

Both Experiments 1 and 2 were programmed on E-Prime 3.0 (Psychology Software Tools, E-Prime^®^, 2018). The procedure is illustrated in Figure 1. Participants were told they would see emotionally neutral or negative pictures and complete a computational estimation task. Participants were informed that, for each problem, the better strategy was either rounding down or rounding up. They were asked to select the better strategy from these two to find the best estimate (or the closest to the correct product) for each problem. Participants were explicitly told that the better strategy was to round one operand down and the other up to their nearest decades (e.g., doing 20 × 70 to estimate 24 × 68) for all problems, but that this mixed-rounding strategy was not allowed. This mixed-rounding strategy was not allowed to make the strategy choice process harder, given that previous studies showed that when participants can use mixed-rounding, strategy selection becomes easy that everybody selects the better strategy on more than 95% of problems (e.g., [39]).

Each trial started with a 1000-ms white screen followed by a 500-ms fixation cross (“*”). Then, an emotionally neutral or negative (i.e., disgusting, fearful, or sad) picture was displayed on the screen for 1500 ms. Following this delay, the multiplication problem appeared superimposed on the picture until the participant’s response. To find out the strategy chosen by the participants, they were asked to calculate and provide their answers out loud. Immediately after the participants provided the response, the experimenter clicked the mouse to record the response time and to trigger the display of a white screen, during which the experimenter recorded the strategy and final response. After a training session of 12 trials, all participants completed 96 trials divided into two blocks of 48 trials.

### 2.2. Results

Results are reported in two main parts. The first part examines age-related differences in how emotions influence participants’ better strategy selection, and the second examines age-related differences in the effects of emotions on participants’ performance (i.e., solution times and percent deviations; see Table 4). Percent deviations (between exact products and participants’ product estimates) were calculated as follows: [(exact product − estimate)/exact product) × 100]. For example, a participant providing 6400 as an estimate for 74 × 89 would be [(6586–6400)/6586] × 100 = 2.8% away from the exact product. Mean percentages of better strategy selection, estimation latencies, and percent deviations were analyzed with mixed-design ANOVAs, 2 (age: young adults, older adults) × 2 (strategy: rounding-down, rounding-up) × 4 (emotion: neutral, disgust, fear, sadness), with repeated measures on the last two factors (see Table 5 for summary of statistical results).

#### 2.2.1. Effects of Emotions on Better Strategy Selection

The main effect of emotions on the percentages of better strategy selection was significant. Participants selected the better strategy less often in the disgust condition (56.3%) compared to the neutral condition (64.7%), *p* < 0.001. In contrast, percentages of better strategy selection in the fear (64.7%) and sadness (66.3%) conditions did not significantly differ from those in the neutral condition, *p* = 0.915 and *p* = 0.465, respectively.

Moreover, the Strategy × Emotion interaction was significant. Participants selected the better strategy less often in the disgust condition than in the neutral condition when the better strategy was the rounding-down strategy (−15.5%), *p* < 0.001, whereas no differences emerged between disgust and neutral conditions when the better strategy was the rounding-up strategy, *p* = 0.647. Participants selected the better strategy more often in the sadness condition than in the neutral condition when the better strategy was the rounding-down strategy (i.e., +5.7%), *p* = 0.027, but equally often in sadness and neutral conditions when the better strategy was the rounding-up strategy, *p* = 0.108. The effects of fear were non-significant in either of the strategy conditions.

In addition, the Age × Emotion interaction was significant. Specifically, young adults selected the better strategy less often in the disgust condition (−11.4%, *p* < 0.001), and more often in the sadness condition (+3.8%, *p* = 0.047), compared to the neutral condition. In contrast, older adults showed no significant effects of disgust, *p* = 0.051, or sadness, *p* = 0.470. For both age groups, fear had no effect (*p* = 0.551 and *p* = 0.259, respectively). No other main or interaction effects were significant for the mean percent selection of the better strategy.

#### 2.2.2. Effects of Emotions on Performance

Young adults were slower than older adults (6999 ms vs. 5432 ms). Participants were slower with the rounding-up strategy compared to the rounding-down strategy (i.e., 6763 ms vs. 5962 ms). They provided poorer estimates with the rounding-up strategy than with the rounding-down strategy (i.e., +1.1% deviation). Moreover, participants were slower in the fear condition than in the neutral condition (+397 ms), *p* = 0.032. However, they were as fast in the disgust and sadness conditions as in the neutral condition (*p* = 0.396 and *p* = 0.329, respectively).

The Age × Strategy interaction revealed that the effect of strategy was significant in young adults (+1143 ms; *p* < 0.001), but not in older adults (*p* = 0.198). Moreover, the Age x Emotion interaction showed that young adults were slower under the fear condition than in the neutral condition (+760 ms), *p* < 0.001. However, they were as fast in the disgust and sadness conditions as in the neutral condition (*p* = 0.466 and *p* = 0.994, respectively). In contrast, older adults showed no significant effects of fear (*p* = 0.568), disgust (*p* = 0.617), or sadness (*p* = 0.204). No other interaction effects were significant for estimation latencies or percentages of deviation.

### 2.3. Summary of Findings

Experiment 1 aimed to determine (a) whether different types of negative emotions differentially influence the ability to select the better strategy for solving arithmetic problems and (b) whether age-related differences in the effects of emotions on strategy selection depend on the type of negative emotions. Young and older adults were asked to choose between the rounding-down and the rounding-up strategies to estimate the products of complex two-digit multiplication problems that were displayed superimposed on emotionally neutral or negative (disgust, fear, or sad) pictures.

The findings showed that not all negative emotions led participants to select the better strategy on each problem less often. Actually, participants selected the better strategy less often, only under disgust, and this happened especially when problems were best estimated with the rounding-down strategy. Moreover, surprisingly, under sadness, participants selected the rounding-down strategy more often when it was the better strategy. Sadness did not change the use of the rounding-up strategy on problems where it was the better strategy.

Concerning performance, fear was the only negative emotion that slowed participants, particularly when the chosen strategy was the rounding-up strategy. This effect of fear on performance was not accompanied by corresponding effects on better strategy selection. Similarly, significant effects of disgust on strategy selection did not lead participants to be slower in providing their estimates. Of course, findings from latency should be interpreted with caution because they may result from different sources, including strategy selection and/or execution. By controlling for strategy selection, Experiment 2 aimed to test the effects of different negative emotions on estimation performance, all else equal.

The present pattern of decreased strategy selection under disgust, increased strategy selection under sadness, and slowed estimation under fear was found primarily in young adults. Older adults’ strategy selection and estimation latencies were not influenced by any negative emotions. These results confirm our hypothesis and replicate previous findings that older adults are less influenced by negative emotions than young adults, regardless of the type of negative emotions. Such findings are easily accounted for by the often-found age-related positivity biases, showing that older adults are less influenced by negative emotions than young adults. Even if our results are clear regarding no effects of emotions on strategy selection, it is hard to conclude that they did not affect performance because age-related differences in strategy performance under different negative emotions may be confounded by age-related differences in strategy selection. In Experiment 2, because all young and older adults executed available strategies on all problems, we could determine whether different emotions influence young and older adults’ performance similarly or differently.

## 3. Experiment 2: Age-Related Differences in Effects of Emotions on Strategy Execution

### 3.1. Method

The same problems (see Table 2) and pictures (see Table 3), as well as the same sequence of events (see Figure 1) within a given trial as in Experiment 1, were used. The only difference was that participants did not have to select strategies on each problem but were told which strategy to execute. The 48 problems were presented twice, distributed across two matched subsets, subset A and subset B. In each subset, half the problems were best estimated with the rounding-down strategy and the other problems with the rounding-up strategy. Half the participants were asked to execute the rounding-down strategy to find estimates of all problems in subset A (once under neutral and once under negative emotion conditions, randomly presented) and to execute the rounding-up strategy on all problems in subset B (also once under neutral and once under negative emotion conditions, randomly presented). The other participants did the reverse. Half the participants executed the rounding-down strategy first and the rounding-up strategy second; the other participants did the reverse. Participants first practiced each strategy on 12 problems, during which the experimenter verified participants’ adherence to the instructed strategies.

### 3.2. Results

Mean estimation times, percentages of correct estimations given the instructed strategy (i.e., if a participant had to execute the rounding-down strategy on 52 × 79 and provided 3500 as an estimate, this participant’s answer was considered correct and coded 1; otherwise, it was coded 0), and mean percent deviations were analyzed with mixed-design ANOVAs, 2 (age: young, older adults) × 2 (strategy: rounding-down, rounding-up) × 4 (emotion: neutral, disgust, fear, sadness), with repeated measures on the last two factors (see Table 6 for means, and Table 7 for summary of statistical results).

#### 3.2.1. Age-Related Differences in Effects of Emotions on Mean Estimation Times

Young adults were slower than older adults (5132 ms vs. 4065 ms). Moreover, participants were slower with the rounding-up strategy than with the rounding-down strategy (5241 ms vs. 3967 ms). Interestingly, the main effect of emotion showed that, relative to the neutral condition, participants were slower in the disgust (+144 ms, *p* = 0.049) and fear (+490 ms, *p* < 0.001) conditions. Moreover, the effect of fear was larger than the effect of disgust, *t*(92) = 3.257, *p* = 0.002. No differences emerged between the neutral and the sadness conditions, *p* = 0.354.

The Strategy × Emotion interaction showed that the effect of disgust was significant when participants executed the rounding-down strategy (+287 ms; *p* < 0.001), but was non-significant when they executed the rounding-up strategy (*p* = 0.996). In contrast, although effects of fear (i.e., fear–neutral) were significant with the rounding-up strategy (+670 ms, *p* < 0.001) or with the rounding-down strategy (+309 ms, *p* < 0.001), these effects were larger for the former than the latter, *t*(92) = 1.998, *p* = 0.049. The effect of sadness was not significant for either the rounding-down strategy (*p* = 0.580) or the rounding-up strategy (*p* = 0.152). Moreover, the Age × Emotion interaction was significant. Young adults showed slowed reaction times under disgust (+286 ms, *p* = 0.005) and fear (+761 ms, *p* < 0.001), but not under sadness (*p* = 0.548). Older adults showed no significant effects of fear (*p* = 0.168), disgust (*p* = 0.993), or sadness (*p* = 0.548).

Finally, the Age × Strategy × Emotion interaction was significant. Post hoc comparison analyses revealed that the effects of emotions differed with each strategy in young adults. Young adults showed larger effects of disgust while executing the rounding-down strategy (+499 ms, *p* < 0.001) than while executing the rounding-up strategy (+74 ms, *p* = 0.607). In contrast, the effects of fear and sadness were larger with the rounding-up strategy (+1107 ms, *p* < 0.001 and +377 ms, *p* = 0.025, respectively) than with the rounding-down strategy (+416 ms, *p* < 0.001 and +8 ms, *p* = 0.931, respectively). Older adults were not slowed down by any emotion, regardless of the strategy they used.

#### 3.2.2. Age-Related Differences in Effects of Emotions on Percentages of Correct Responses and Percentages of Deviations

Analyses of percent correct responses showed that older adults were more accurate than young adults (92.3% vs. 88.5%). Participants were more accurate with the rounding-down strategy than with the rounding-up strategy (92.2% vs. 87.9%). Moreover, the main effects of emotions showed that participants were less accurate in the fear condition than in the neutral condition (−2.7%), *p* = 0.009, but were as accurate in the disgust and sadness conditions as in the neutral condition, *p* = 0.210 and *p* = 0.578, respectively. Finally, the Strategy × Emotion revealed that participants were less accurate under fear only while executing the rounding-up strategy (−3.8%), *p* = 0.011. Furthermore, participants were more accurate under disgust while executing the rounding-up strategy (+3.1%), *p* = 0.010, but not while executing the rounding-down strategy (−0.6%), *p* = 0.689. Finally, regardless of the strategy, sadness did not affect accuracy.

Interestingly, the main effect of emotions also came out significant in percent deviations. Participants provided poorer estimates in the disgust condition than in the neutral condition (+0.9% deviation, *p* = 0.002). On the contrary, participants provided better estimates in the fear condition than in the neutral condition (−0.9% deviation, *p* < 0.001). This indicates that although participants committed estimation errors more frequently under fear, the magnitude of these errors was smaller than in the neutral condition. Conversely, while the frequency of errors under disgust did not differ from that observed in the neutral condition, the magnitude of these errors was greater in the disgust condition. As for the percentages of correct responses, sadness had no effect on percent deviations. No other interaction effects came out significant on percentages of correct responses and on percent deviations.

### 3.3. Summary of Findings

Experiment 2 aimed at determining (a) whether different types of negative emotions had the same influence on the execution of strategies while solving arithmetic problems and (b) whether age-related differences in the effects of negative emotions on strategy execution depend on the type of negative emotions. Unique to Experiment 2 was that we controlled all other strategy dimensions (i.e., strategy repertoire, distribution, and selection). The results support our first hypothesis that different types of negative emotion differentially influence the speed of strategy execution in arithmetic. Disgust and fear led participants to be slower to execute instructed strategies, with a greater slowdown under fear, whereas sadness did not affect latencies. Fear also led to decreased accuracy. Most interesting, the deleterious effects of disgust and fear were modulated by the instructed strategies. Thus, the effects of disgust were larger while participants executed the rounding-down strategy. This was unexpected because the rounding-up strategy is typically more difficult and more resource-demanding than the rounding-down strategy. If negative emotions disrupt performance by capturing attentional resources, their effects should be stronger for the more demanding rounding-up strategy. In this study, this pattern emerged only for fear, whose deleterious effect on strategy execution latencies and accuracy was indeed greater for the rounding-up strategy.

When comparing the two age groups separately, this pattern of slower strategy execution under disgust (particularly for the rounding-down strategy) and fear (particularly for the rounding-up strategy) appeared only in young adults. Older adults’ strategy execution was not influenced by any negative emotions, regardless of which strategy they were asked to implement. These results on strategy execution replicate those on strategy selection in Experiment 2 and show that the target negative emotions (fear, disgust, sadness) did not affect the efficiency with which older adults executed strategies.

## 4. General Discussion

The present study aimed to determine (a) whether different types of negative emotion differentially influence strategy selection and strategy execution while solving arithmetic problems, and (b) whether age-related differences in the effects of negative emotions depend on the type of emotion. Young and older adults were asked to estimate the products of two-digit complex multiplication problems displayed on emotionally neutral or negative (disgust, fear, sad) pictures by choosing between the rounding-down and the rounding-up strategies (Experiment 1, assessing strategy selection) or by executing instructed strategies (Experiment 2, assessing strategy execution). We found that not all negative emotions influence arithmetic performance and that young and older adults were not influenced the same way by negative emotions.

### 4.1. How Several Types of Negative Emotions Influence Strategic Aspects of Arithmetic Performance

The present results of deleterious effects of negative emotions in arithmetic replicate previous findings (e.g., [12]; [15], [16]; [19], [20]; [28], [29]; [33], [34]; [35]; [38]; [42]; [45], [46]; [61]; [67], [68], [69]) and confirmed that these deleterious effects are mediated by disruptions of strategic aspects of arithmetic performance (e.g., [19]; [35]; [38]). The first key original finding of this study is that different types of negative emotions do not have the same influence on arithmetic performance.

These results support our first hypothesis that fear, sadness, and disgust have different effects on strategic aspects of arithmetic performance. Specifically, here, we found that disgust led participants to select the better strategy on each problem less often and to execute the instructed strategies more poorly. Fear had no effect on strategy selection but slowed the execution of strategies to a greater extent than disgust. Our findings also showed that sadness has no influence on either strategy selection or execution. Although these findings support the general hypothesis that discrete negative emotions exert different effects, they are not consistent with our more specific predictions derived from Pekrun’s Cognitive Value Theory (CVT; [49], [50]; [51]). The CVT predicted decreased performance under deactivating negative emotions (sadness) and greater engagement in the task and/or disengagement from activating negative emotions (disgust, fear). Our results, therefore, diverge from these theoretical expectations, suggesting that the activating/deactivating distinction may not account for the emotion-specific patterns observed here.

How do different negative emotions exert their effects on arithmetic performance? Differences in valence and intensity across these emotions can be ruled out, as valence and arousal did not differ across disgust, fear, and sadness.

The functional significance of emotions may explain how they differ in their effects on arithmetic performance. Indeed, participants’ attitudes toward emotions may vary with the type of emotions ([22]). Disgust and fear are highly salient emotions, as they enable individuals to avoid potential contamination or aversive situations and to quickly respond to potential threats. As a consequence, they may exert strong attentional capture, which could lead to their disruptive influence here. The greater effect of fear compared with disgust on strategic aspects of performance may stem from the fact that detecting potential danger is even more critical for survival than identifying an aversive but non-threatening situation. In contrast, sadness appears to be a less salient emotion, whose detection is less directly tied to survival. Sadness typically arises in response to negative events such as loss and is often accompanied by a motivation to change the situation. It is therefore plausible that the sad pictures captured participants’ attention to a lesser extent, or that participants focused more on the arithmetic task to regulate the sadness elicited by pictures, leading to unaffected strategy selection and execution.

Alternatively, different types of negative emotions may exert their effects on different mechanisms involved in estimating multiplication problems. For example, using magnetoencephalography and a dimensional approach of emotions, [35] ([35]) found that negative emotions exerted their deleterious effects on execution strategy by specifically disrupting the encoding of arithmetic problems. Future studies could aim to determine which mechanisms are specifically disrupted by disgust, fear, and sadness. Moreover, a limitation of the present study is the absence of self-report assessments of the emotional experience elicited by the pictures. Future studies should include participants’ ratings of valence and arousal, as well as categorical emotion judgments for each picture, to ensure that the stimuli effectively induce the intended emotions and to better account for potential individual differences in emotional reactivity.

### 4.2. Age-Related Changes in the Effects of Negative Emotions on Strategy Aspects of Arithmetic Performance

The second key finding of the present study is that age-related differences in the effects of negative emotions on strategic aspects of arithmetic performance do not depend on the type of negative emotions. These findings support our second general hypothesis of age-related differences in the effects of negative emotions on strategic aspects of arithmetic performance, and our more precise hypothesis that older adults are less influenced by negative emotions than young adults. Older adults were less influenced than young adults by negative emotions, both when choosing between the rounding-down and the rounding-up strategies and when executing the instructed rounding strategies to estimate the products of multiplication problems.

These findings are indeed consistent with previous research documenting robust age-related positivity effects across a variety of cognitive domains (see [3]; [5]; [44]; [54]; [60], for reviews) and extends more recent evidence of such age-related positivity effects in arithmetic (e.g., [33]; [38]).

More specifically, our data replicates [38]’s ([38]) finding that negative emotions have a smaller influence on strategy execution in older adults than in young adults. We also found reduced emotion effects on older adults’ strategy selection relative to young adults. This finding contrasts with Lemaire’s finding of no age differences in emotion effects on strategy selection. Several methodological factors may account for the divergence. For example, the pictures used in the two studies were not strictly identical. In addition, while Lemaire adopted a dimensional approach to emotions, contrasting a neutral condition with a single, general negative condition, the present study adopted a typological approach to emotions, in which one-third of the pictures induced fear, disgust, or sadness. A plausible explanation for the discrepancy is that a dimensional analysis of negative emotions may mask age-related subtleties. When negative emotions are collapsed into a single negative valence measure, overall effects may appear comparable among young and older adults. In contrast, a more fine-grained typological approach of emotions can reveal that different negative emotions, such as disgust, fear, and sadness, exert a weaker influence on strategy selection in older adults. Future research would therefore benefit from directly comparing dimensional and typological frameworks within the same experimental protocol, allowing researchers to disentangle global valence effects from emotion-specific patterns and to clarify how aging moderates effects of emotion in arithmetic.

Smaller effects of negative emotions on arithmetic performance may arise from older adults’ attention being less captured by negative stimuli than that of young adults, as found in prior studies on attention (e.g., [1]; [6]; [9]; [18]; [25], [26]; [24]; [30]; [43]; [47]; [57]; [59]; see [23]; [58], for reviews). The present findings highlight that this reduced attentional capture by negative emotions extends across different types of negative emotions. Older adults’ attention may have been less captured by fear, disgust, or sadness, leaving their attentional and cognitive resources available to select the more efficient strategy or to execute the instructed strategies more efficiently to solve arithmetic problems.

An alternative explanation is that both young and older adults experienced similar attentional capture by negative pictures, but older adults regulated these emotional responses more effectively, consistent with evidence that older adults are sometimes better than young adults at regulating irrelevant negative emotions (see [2]; [4]; [10]; [55], for reviews). From this perspective, older adults would be better able to downregulate irrelevant negative emotions induced by pictures, regardless of specific emotional category. Future research could help disentangle these possibilities by directly assessing attentional capture and emotion-regulation processes in the same paradigm.

A further explanation that cannot totally be ruled out, and represents a potential limitation of the present study, is that older adults were generally more skilled at arithmetic than young adults, as indicated by their higher scores on the French-Kit and their superior task performance. Across emotional conditions, older adults were faster and more accurate than young adults when executing the required strategies. Such greater baseline expertise may have acted as a protective factor against the detrimental effects of negative emotions. In addition, although both young and older adults obtained comparable arithmetic fluency scores in our paper-and-pencil French Kit test, it is possible that older adults relied more heavily on automatized arithmetic knowledge, such as well-consolidated multiplication facts, which could reduce the cognitive demands associated with strategy selection and execution and thereby reduce emotional interference. However, this baseline difference between young and older adults was observed only in the second experiment. In the first experiment, where young and older adults had similar baseline performance, positivity biases were still observed. Finally, it should be noted that the older adults included in the present study were cognitively and physically active volunteers. How the present findings generalize across different older individuals should be tested in future research.

## 5. Conclusions

In the present study, young and older adults performed arithmetic tasks in which they had to select or execute the most appropriate strategy to solve arithmetic problems under different emotional conditions. We examined how fear, disgust, and sadness influence strategic aspects of arithmetic performance and whether these effects differ between young and older adults. Taken together, the findings of the present study carry both empirical and theoretical implications.

Empirically, our findings demonstrate that discrete negative emotions exert different effects on arithmetic performance. Indeed, fear and disgust, but not sadness, impaired arithmetic performance. A second empirical implication concerns age-related differences in the effects of emotions on cognition. Older adults were less influenced by negative emotions such as fear, disgust, or sadness. This finding extends age-related positivity effects to several discrete negative emotions and helps clarify the conditions under which reduced emotional interference emerges in later adulthood.

Theoretically, the present results contribute to models of emotion–cognition interactions by suggesting that the effects of different negative emotions on arithmetic performance operate through the strategies participants are using to perform the target cognitive task. Specifically, discrete negative emotions modulate the ability to select the most efficient strategies and to execute them effectively. However, several theoretical challenges remain. First, our data did not enable us to determine the loci of effects of each emotion, as different emotions may influence different processes in target cognitive tasks in general and arithmetic (e.g., encoding problems, calculating correct answers, responding) in particular. Future studies may further our understanding of how emotions influence arithmetic performance by determining which processes of the target tasks emotions influence. Also, the lack of effects of fear, disgust, and sadness in older adults does not mean that they are not influenced by any negative emotions. Future studies testing other negative emotions may find evidence of their influence on arithmetic performance in older adults.

## Figures and Tables

**Figure 1 behavsci-15-01695-f001:**
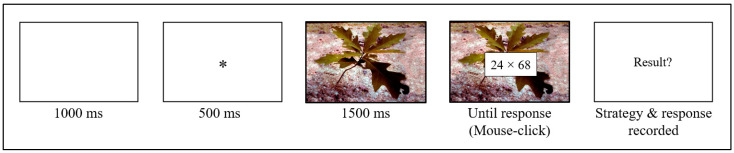
Sequence of events within an example of an emotionally neutral trial.

**Table 1 behavsci-15-01695-t001:** Participants’ characteristics in Experiments 1 and 2.

Characteristics	Young Adults	Older Adults	*Fs*
Experiment 1. Strategy selection			
*N* (females/males)	39 (29/10)	27 (20/7)	-
*Mean* age in y.m. (*SD*)	21.1 (3.2)	70.3 (5.5)	-
*Range*	18–32	65–89	-
*Mean* number of years of formal education (*SD*)	14.3 (0.8)	14.0 (2.0)	0.51
Arithmetic fluency (*SD*)	39.0 (12.1)	57.6 (14.9)	31.11 ***
Vocabulary (*SD*)	20.7 (3.9)	26.3 (3.7)	33.32 ***
MMSE (*SD*)	-	28.7 (1.0)	-
Experiment 2. Strategy execution			
*N* (females/males)	47 (29/18)	46 (26/20)	-
*Mean* age in y.m. (*SD*)	21.3 (3.0)	71.3 (5.0)	-
*Range*	18–31	65–86	-
*Mean* number of years of formal education (*SD*)	13.7 (1.3)	14.4 (3.3)	1.44
Arithmetic fluency (*SD*)	40.3 (16.2)	59.2 (18.7)	27.15 ***
Vocabulary (*SD*)	20.1 (4.6)	26.0 (3.6)	46.64 ***
MMSE (*SD*)	-	28.9 (1.0)	-

Note. *** *p* < 0.001. Arithmetic fluency was assessed using a paper-and-pencil arithmetic test ([17]) in which participants had to solve as many addition, subtraction, and multiplication problems as possible in six minutes. Vocabulary was assessed using a French version of the Mill-Hill Vocabulary Scale (MHVS; [8]; [53]). MHVS consists of 34 items distributed across three pages. Each item included a target word followed by six proposed words, and the task consisted of identifying which word was the closest to the target. The French version of the Mini-Mental-Status Examination (MMSE; [14]) assessed general cognitive abilities. None of the participants had a score below 27. All participants had corrected vision, and all stated that they had no hearing problems.

**Table 2 behavsci-15-01695-t002:** List of multiplication problems used in both Experiments 1 and 2.

RD Problems		RU Problems	
43 × 79	71 × 57	34 × 49	68 × 73
43 × 86	71 × 58	34 × 59	69 × 54
46 × 83	72 × 46	48 × 71	72 × 49
47 × 51	72 × 47	49 × 82	73 × 58
51 × 87	76 × 51	53 × 79	74 × 89
52 × 79	76 × 81	54 × 68	79 × 54
57 × 61	79 × 42	58 × 63	79 × 63
57 × 72	79 × 62	59 × 63	79 × 64
61 × 76	81 × 46	59 × 74	83 × 59
63 × 86	82 × 57	62 × 59	84 × 47
68 × 71	86 × 53	64 × 87	84 × 48
69 × 52	87 × 52	68 × 53	87 × 74

**Table 3 behavsci-15-01695-t003:** IAPS references of pictures used in the present study ([36]).

Neutral Pictures	Negative Pictures		
2038, 2190, 2393, 2397, 2397, 2411, 2440, 2480, 2570, 2840, 2850, 2880, 2890, 5510, 5520, 5530, 5740, 7000, 7004, 7006, 7010, 7012, 7020, 7025, 7026, 7030, 7031, 7035, 7040, 7041, 7050, 7053, 7059, 7080, 7090, 7100, 7110, 7150, 7161, 7175, 7179, 7185, 7187, 7217, 7235, 7491, 7705, 7950	Disgust	Fear	Sadness
	2730, 2981, 3019, 3103, 3140, 3250, 3550, 6415, 9042, 9300, 9302, 9321, 9373, 9400, 9405, 9500	1090, 1111, 1201, 1202, 1205, 1220, 3500, 3530, 6242, 6300, 6510, 6520, 6821, 6825, 6831, 6838	2301, 2688, 3300, 6570.1, 9002, 9050, 9184, 9250, 9900, 9902, 9903, 9904, 9905, 9910, 9911, 9920

**Table 4 behavsci-15-01695-t004:** Better strategy selection (%), estimation latencies (ms), and percentages of deviations as a function of age (young, older adults), strategy (rounding-down, rounding-up), and emotion (neutral, disgust, fear, sadness) in Experiment 1.

	Young Adults (*N* = 39)							Older Adults (*N* = 27)						
	Neutral	Disgust	Fear	Sadness	Disgust–Neutral	Fear–Neutral	Sadness–Neutral	Neutral	Disgust	Fear	Sadness	Disgust–Neutral	Fear–Neutral	Sadness–Neutral
Better strategy selection (%)														
Rounding-Down	66.1	46.4	63.9	74.1	−19.8 ***	−2.2	8.0 **	59.7	50.2	56.0	62.1	−9.4 **	−3.7	2.4
Rounding-Up	65.9	62.7	66.0	65.5	−3.1	0.1	−0.4	66.1	67.2	72.5	60.3	1.1	6.4 *	−5.7 *
*Means*	66.0	54.5	64.9	69.8	−11.4 ***	−1.0	3.8 *	62.9	58.7	64.2	61.2	−4.2	1.3	−1.6
Estimation latencies (ms)														
Rounding-Down	6267	6307	6902	6233	40	636 *	−34	5435	5331	5041	5351	−104	−394	−84
Rounding-Up	7423	7091	8307	7459	−332	884 ***	36	5681	5545	5820	5339	−136	139	−342
*Means*	6845	6699	7605	6846	−146	760 ***	1	5558	5438	5430	5345	−120	−128	−213
Percentages of deviations														
Rounding-Down	4.5	4.5	4.8	6.7	−0.1	0.2	2.2 **	5.1	5.6	5.2	6.9	0.5	0.1	1.8
Rounding-Up	6.7	6.7	8.7	6.1	−0.1	2.0 *	−0.7	4.7	7.2	6.0	5.7	2.5 *	1.4	1.1
*Means*	5.6	5.6	6.8	6.4	−0.1	1.1	0.8	4.9	6.4	5.6	6.3	1.5	0.7	1.4 *

Note. * *p* < 0.05; ** *p* < 0.01; *** *p* < 0.001. Better strategy selection. Young adults selected the better strategy (especially the RD strategy) less often in the disgust condition and more often in the sadness condition, compared to the neutral condition. Older adults showed no main effects of negative emotions. Estimation latencies. Young adults were slower in the fear condition than in the neutral condition. Older adults showed no main effects of negative emotions.

**Table 5 behavsci-15-01695-t005:** Statistics of effects on percentages of better strategy selection, estimation latencies, and percentages of deviations.

Effects	*MSe*	*Fs*	*p*	*η^2^p*
Better strategy selection (%)				
Age	1103.403	0.492	0.485	0.008
Strategy	1115.703	3.936	0.052	0.058
Emotion	189.137	12.808	<0.001	0.167
Age × Strategy	1115.703	1.393	0.242	0.021
Age × Emotion	189.137	5.253	0.002	0.076
Strategy × Emotion	156.848	17.502	<0.001	0.215
Age x Strategy × Emotion	156.848	1.630	0.184	0.025
Estimation latencies				
Age	35,002,964.210	8.826	0.004	0.121
Strategy	2,998,814.940	22.365	<0.001	0.259
Emotion	2,507,559.331	2.979	0.050	0.044
Age × Strategy	2,998,814.940	7.440	0.008	0.104
Age × Emotion	2,507,559.331	3.185	0.040	0.047
Strategy × Emotion	1,943,481.481	1.248	0.294	0.019
Age × Strategy × Emotion	1,943,481.481	0.440	0.700	0.007
Percentages of deviations				
Age	26.431	0.428	0.515	0.007
Strategy	27.744	5.214	0.026	0.075
Emotion	18.851	1.705	0.172	0.026
Age × Strategy	27.744	3.494	0.066	0.052
Age × Emotion	18.851	1.358	0.259	0.021
Strategy × Emotion	12.116	5.954	0.001	0.085
Age x Strategy × Emotion	11.618	1.267	0.287	0.019

**Table 6 behavsci-15-01695-t006:** Estimation latencies (ms), percentages of correct estimations, and percentages of deviations as a function of age (young, older adults), strategy (rounding-down, rounding-up), and emotion (neutral, disgust, fear, sadness), in Experiment 2.

	Young Adults (*N* = 47)							Older Adults (*N* = 46)						
	Neutral	Disgust	Fear	Sadness	Disgust–Neutral	Fear–Neutral	Sadness–Neutral	Neutral	Disgust	Fear	Sadness	Disgust–Neutral	Fear–Neutral	Sadness–Neutral
Estimation latencies (ms)														
Rounding-Down	4197	4696	4613	4205	499 ***	416 ***	8	3450	3520	3650	3365	71	201	−85
Rounding-Up	5447	5520	6554	5824	74	1107 ***	377 *	4605	4532	4829	4568	−73	224	−37
*Means*	4822	5108	5583	5014	286 **	761 ***	193	4027	4026	4239	3967	−1	212	−61
Percentages of correct estimations														
Rounding-Down	91.3	90.4	88.7	92.3	−0.9	−2.6	1.0	96.0	95.7	95.2	93.9	−0.3	−0.8	−2.1
Rounding-Up	86.4	90.0	81.4	87.5	3.6 *	−5.0 *	1.1	89.0	91.6	86.4	90.8	2.6	−2.6	1.8
*Means*	88.9	90.2	85.1	89.9	1.3	−3.8 **	1.1	92.5	93.6	90.8	92.4	1.2	−1.7	−0.1
Percentages of deviations														
Rounding-Down	15.7	16.7	15.5	15.5	1.0 *	−0.2	−0.2	15.9	16.3	14.6	15.6	0.4	−1.3 ***	−0.3
Rounding-Up	16.3	17.9	15.2	15.8	1.6 **	−1.1 *	−0.5	15.9	16.8	15.0	16.3	0.9	−0.9	0.4
*Means*	16.0	17.3	15.3	15.7	1.3 **	−0.7 *	−0.3	15.9	16.6	14.8	16.0	0.7	−1.1 ***	0.1

Note. * *p* < 0.05; ** *p* < 0.01; *** *p* < 0.001. Estimation latencies. Young adults were slower under disgust and under fear than in the neutral condition. Older adults showed no effects of negative emotions. Percentages of correct estimations. Young adults were less accurate under fear than in the neutral condition. Older adults showed no effects of negative emotions. Percentages of deviations. Young adults provided poorer estimates in the disgust condition than in the neutral condition and provided better estimates in the fear condition than in the neutral condition. Older adults also provided better estimates in the fear condition than in the neutral condition.

**Table 7 behavsci-15-01695-t007:** Statistics of effects on estimation latencies, percentages of correct responses, and percentages of deviations.

Effects	*MSe*	*Fs*	*p*	*η^2^p*
Estimation latencies				
Age	19,868,106.74	10.657	0.002	0.105
Strategy	2,689,218.149	112.065	<0.001	0.552
Emotion	2,406,205.535	10.877	<0.001	0.107
Age × Strategy	2,689,218.149	1.271	0.263	0.014
Age × Emotion	2,406,205.535	2.928	0.043	0.031
Strategy × Emotion	847,725.522	5.400	0.003	0.056
Age x Strategy × Emotion	847,725.522	2.931	0.047	0.031
Percentages of correct responses				
Age	557.271	4.863	0.030	0.051
Strategy	229.846	20.428	<0.001	0.183
Emotion	107.263	5.653	<0.001	0.058
Age × Strategy	229.846	0.379	0.540	0.004
Age × Emotion	107.263	0.905	0.432	0.010
Strategy × Emotion	95.989	3.567	0.020	0.038
Age x Strategy × Emotion	95.989	0.683	0.542	0.007
Percentages of deviations				
Age	9.780	1.255	0.265	0.014
Strategy	9.372	3.447	0.067	0.036
Emotion	7.610	17.112	<0.001	0.158
Age × Strategy	9.372	0.024	0.878	0.000
Age × Emotion	7.610	1.667	0.183	0.018
Strategy × Emotion	6.674	0.784	0.481	0.009
Age × Strategy × Emotion	6.674	0.888	0.431	0.010

## Data Availability

Data can be found on the following link: https://osf.io/zcpx8/overview?view_only=324bf69ba0f64760a2b5f30984b060b2 accessed on 21 October 2025.

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
