# Peer review of "Age-Related Differences in How Fear, Disgust, and Sadness Influence Strategic Aspects of Arithmetic Performance"

_behavsci, 2025, doi:10.3390/bs15121695_

Round 1
Reviewer 1 Report
Comments and Suggestions for Authors
This work makes an important contribution to the understanding of the role of emotions in the performance of arithmetic tasks. It uses a methodology based on a discrete or categorical conception of emotions. From a methodological standpoint, I consider it well done. I suggest the following changes to potentially improve the work:
- The abstract begins by directly describing the methodology. It would be advisable to briefly introduce a description of the context or background, as well as the research objective(s) and question(s).
-In the final part of the procedure description, it is unclear whether, once the participants verbalized the result of their calculation, they themselves were responsible for clicking the mouse to record the response time and display the next white screen. If so, it should be noted that the participants had been previously instructed to click the mouse immediately after giving their verbal response. It seems to be implied that the participants themselves, not the experimenter, were responsible for clicking the mouse, but it is important to clarify this to facilitate the possible replication of the experiment, should the need arise.
-In section 4.2, possible explanations are presented as to why older adults are less influenced by negative emotions in the arithmetic task. Perhaps it could be considered that older adults have greater attentional capacities than younger adults, or even, related to this, a greater degree of automatization of some arithmetic tasks, such as the retrieval of numerical facts. It might be interesting to add some possible reference to such possibilities, should such possibilities exist.
Author Response
- The abstract begins by directly describing the methodology. It would be advisable to briefly introduce a description of the context or background, as well as the research objective(s) and question(s).
The abstract has now been revised to include a brief contextualization of the research problem, and a clear presentation of the research objectives and research questions (p.1).
- In the final part of the procedure description, it is unclear whether, once the participants verbalized the result of their calculation, they themselves were responsible for clicking the mouse to record the response time and display the next white screen. If so, it should be noted that the participants had been previously instructed to click the mouse immediately after giving their verbal response. It seems to be implied that the participants themselves, not the experimenter, were responsible for clicking the mouse, but it is important to clarify this to facilitate the possible replication of the experiment, should the need arise.
Thank you for this valuable comment, which indeed helps clarify our procedure. The experimenter, not the participants, was responsible for clicking the mouse to record the response time. We have now explicitly added this information to the procedure section (p. 8, 271-272) to ensure full transparency and replicability of the experiment.
- In section 4.2, possible explanations are presented as to why older adults are less influenced by negative emotions in the arithmetic task. Perhaps it could be considered that older adults have greater attentional capacities than younger adults, or even, related to this, a greater degree of automatization of some arithmetic tasks, such as the retrieval of numerical facts. It might be interesting to add some possible reference to such possibilities, should such possibilities exist.
This comment aligns closely with one of the interpretations we had already begun to develop regarding age-related differences in the Discussion session. In the revised manuscript, we have expanded this section of the Discussion (p.18, 638-642) to consider more explicitly the possibility that older adults’ reduced sensitivity to negative emotions may stem from greater arithmetic expertise and a higher degree of automatization of some numerical procedures, which could free attentional resources and reduce emotional interference. We are grateful for this insight, which has strengthened our discussion of age-related effects.
Reviewer 2 Report
Comments and Suggestions for Authors
It is imperative to adjust the abstract in terms of introduction, method, main results and discussion.
It is imperative that the introduction incorporates citations to the consulted papers.
It is imperative that the design of the experiment is presented, as opposed to that of the quantitative analysis.
It has been established that there is an inaccuracy in the page counter, with the reading being erroneously elevated after page 9 out of a total of 24 pages.
It is imperative that Table 6 is indexed to the left-hand side of the page. The preceding page contains a substantial blank space. The content should be adapted in order to fill this space.

Author Response
- It is imperative to adjust the abstract in terms of introduction, method, main results and discussion.
The abstract has been restructured to align with IMRaD conventions, providing a concise introduction, a clearer description of the methodological approach, a focused summary of the key results, and a more synthetic concluding statement (p.1).
- It is imperative that the introduction incorporates citations to the consulted papers.
References have been added to the introduction (p. 1).
- It is imperative that the design of the experiment is presented, as opposed to that of the quantitative analysis.
We are not sure what is meant here. Indeed, we detailed the design of our statistical analyses for each reported analysis. How the experiments were designed is described in detail in methods sections of each experiment.
- It has been established that there is an inaccuracy in the page counter, with the reading being erroneously elevated after page 9 out of a total of 24 pages.
The issue with the page numbering has been solved.
- It is imperative that Table 6 is indexed to the left-hand side of the page. The preceding page contains a substantial blank space. The content should be adapted in order to fill this space.
The content has been adapted to fill all the spaces in the entire manuscript.
Reviewer 3 Report
Comments and Suggestions for Authors
Congratulate the authors on their work. The study was carefully planned, is methodologically rigorous, and addresses an important gap in the literature on emotional cognition and aging. By combining a typological approach to negative emotions with a strategy-based arithmetic framework, the authors offer a valuable and original contribution to the field. The experiments were executed with precision, and the statistical analysis is detailed and transparent. The clarity of the tables and the organization of the results are also commendable.
Here are some suggestions to further improve the manuscript.
- Abstract
The abstract is informative but overly descriptive and does not follow IMRAD logic. It would be beneficial to include:
- a brief contextualization of the research problem;
- a more concise description of the methodological approach;
- a clearer synthesis of the main conclusions and implications.
A more concise abstract would improve readability and impact.
- Introduction
The introduction is rich but excessively long, with some theoretical redundancies. Several sections—particularly those on ageing, positivity effects, and strategy frameworks—repeat well-established concepts and could be condensed. A more explicit and initial focus on the specific research gap would improve clarity. The hypotheses would also benefit from clearer and more concise formulation. The hypotheses are mentioned but could be stated more explicitly and end in the Introduction. Presenting them in a clear, concise manner—preferably grouped and immediately linked to the theoretical rationale—would improve the structure and prepare the reader for the subsequent methodological design.
- Methodology
The methodological description is generally robust, but some aspects deserve clarification or improvement:
3.1. Emotional Induction
Although the use of IAPS images is appropriate, the manuscript could better address:
- the limitations of using 1500 ms exposures to induce discrete emotions,
- whether any manipulation checks (self-report assessments) were performed or considered,
- the potential influence of individual differences in emotional reactivity.
Even if no manipulation checks were collected (which is acceptable), explicitly acknowledging this limitation would improve transparency.
3.2. Strategy Instruction (Experiment 2)
The procedure for Experiment 2 is well described, but it may be helpful to clarify:
- whether the order of the instructed strategies might interact with the emotional effects,
- whether participants' adherence to the instructed strategy was monitored beyond accuracy scores.
A brief justification for why the mixed-rounds strategy was prohibited may also help readers generally unfamiliar with previous work.
3.3. Sample Characterization
The sample of elderly individuals was well-selected (MMSE ≥ 27), but:
- a brief statement on the adequacy of hearing/vision could reinforce methodological robustness,
- and a note on potential sampling bias (e.g., elderly volunteers and cognitively active individuals) could be useful in the limitations section.
- Results
4.1. Linking Results to Hypotheses More Explicitly
In several places, the results are presented clearly, but are not explicitly linked to the formulated hypotheses.
4.2. Presentation of the Tables
The tables are extremely detailed (which is positive), but including brief, concise summaries before or after them (e.g., main patterns or most significant contrasts) could facilitate understanding of the results, especially for readers less familiar with strategy-based arithmetic paradigms.
- Discussion
While the discussion addresses the main findings, greater engagement with theoretical frameworks such as Cognitive Value Theory, approach-avoidance models, and age-related compensatory mechanism theories would strengthen the interpretative depth. The absence of emotional effects in older adults is consistent with positive effects, but alternative explanations deserve to be explored.
6 Conclusion and Implications
The conclusion could be strengthened by a clearer synthesis of the practical and theoretical implications. For example, the manuscript could address how distinct emotions differentially affect the strategic components of cognition, or how age-related resilience to negative emotional interference can underpin models of healthy cognitive aging and future research on strategic behavior.
Overall, this is a solid and promising manuscript. With the proposed revisions, the article will provide an even more impactful contribution to the literature.
Author Response
Abstract : The abstract is informative but overly descriptive and does not follow IMRAD logic. It would be beneficial to include:
-a brief contextualization of the research problem;
-a more concise description of the methodological approach;
-a clearer synthesis of the main conclusions and implications.
A more concise abstract would improve readability and impact.
The abstract has been revised to follow IMRAD logic and to incorporate a short contextual introduction, a more concise presentation of the methodological framework, and a clearer synthesis of the main conclusions and implications (p.1).
Introduction: The introduction is rich but excessively long, with some theoretical redundancies. Several sections—particularly those on ageing, positivity effects, and strategy frameworks—repeat well-established concepts and could be condensed. A more explicit and initial focus on the specific research gap would improve clarity. The hypotheses would also benefit from clearer and more concise formulation. The hypotheses are mentioned but could be stated more explicitly and end in the Introduction. Presenting them in a clear, concise manner—preferably grouped and immediately linked to the theoretical rationale—would improve the structure and prepare the reader for the subsequent methodological design.
The introduction has been shortened to focus on the essential theoretical elements and to highlight more clearly the specific gaps in the literature that our study addresses. Redundant sections have been condensed to improve clarity and conciseness. Additionally, the hypotheses are now presented in a clearer and more explicit manner at the end of the Introduction.
Methodology: The methodological description is generally robust, but some aspects deserve clarification or improvement:
Emotional Induction: Although the use of IAPS images is appropriate, the manuscript could better address:
- the limitations of using 1500 ms exposures to induce discrete emotions,
- whether any manipulation checks (self-report assessments) were performed or considered,
- the potential influence of individual differences in emotional reactivity.
Even if no manipulation checks were collected (which is acceptable), explicitly acknowledging this limitation would improve transparency.
We thank the reviewer for these insightful comments.
The 1500-ms exposure duration follows our previous studies using the same type of stimuli and showing that this exposure duration reliably elicited robust emotional effects. In the present study, it also appears sufficient to induce discrete emotions, as evidenced by the differential effects observed across emotion categories.
We agree, however, that including self-report assessments (or physiological and/or eye-movement measures) would have strengthened the design. Such assessments would help verify that emotions were induced and help account for individual differences in emotional reactivity. We have added this methodological limitation in the Discussion section (p. 16, 572-576).
Strategy Instruction (Experiment 2): The procedure for Experiment 2 is well described, but it may be helpful to clarify:
- whether the order of the instructed strategies might interact with the emotional effects,
- whether participants' adherence to the instructed strategy was monitored beyond accuracy scores.
- A brief justification for why the mixed-rounds strategy was prohibited may also help readers generally unfamiliar with previous work.
First, we have now clarified in the manuscript that the order of the strategy instructions was counterbalanced across participants to avoid any interaction between strategy order and effect of emotions (p.11, 392-394). Note that we also conducted additional analyses to test for potential block-order effects, and none were observed.
Second, regarding adherence to the instructed strategies, we consider accuracy scores to provide a reliable indication that participants correctly applied the required procedures. In the neutral condition, accuracy reached 93.6% for the rounding-down strategy and 87.7% for the rounding-up strategy, which confirms that participants understood and executed both strategies appropriately. As detailed in the Method section, participants also completed 12 training trials for each strategy. We have now added that the experimenter verified participants’ adherence to the instructed strategies (p. 11, 394-395).
Finally, concerning the prohibition of the mixed-rounds strategy, this choice is in the Method section in Experiment 1 (p. 7, 259-265): “This mixed-rounding strategy was not allowed to make the strategy choice process harder, given that previous studies showed that when participants can use mixed-rounding, strategy selection is so easy that everybody selects the better strategy on more than 95% of problems (e.g., Lemaire et al., 2004).”
Sample Characterization: The sample of elderly individuals was well-selected (MMSE ≥ 27), but:
- a brief statement on the adequacy of hearing/vision could reinforce methodological robustness,
- and a note on potential sampling bias (e.g., elderly volunteers and cognitively active individuals) could be useful in the limitations section.
In the revised manuscript, we added a sentence to Table 1. (participant’s characteristics) stating that “all participants had corrected vision, and all stated that they had no hearing problems” (p.5, 204-205).
Concerning the potential sampling bias, we added a sentence at the end of the Discussion section (p.17, 645-648): “Finally, it should be noted that the older adults included in the present study were cognitively and physically active volunteers. How the present findings generalize across different older individuals should be tested in future research”.
Results:
Linking Results to Hypotheses More Explicitly: In several places, the results are presented clearly but are not explicitly linked to the formulated hypotheses.
The results are now explicitly linked to the two hypotheses we formulated at the end of the introduction, whenever possible, in the Discussion section (e.g., p.15, 535-548, and p. 16, 581-584).
Presentation of the Tables: The tables are extremely detailed (which is positive), but including brief, concise summaries before or after them (e.g., main patterns or most significant contrasts) could facilitate understanding of the results, especially for readers less familiar with strategy-based arithmetic paradigms.
The tables of results now include a brief description of the main and noteworthy results in their captions (Table 4, p.9, 317-320, and Table 6, p.12, 409-413).
Discussion: While the discussion addresses the main findings, greater engagement with theoretical frameworks such as Cognitive Value Theory, approach-avoidance models, and age-related compensatory mechanism theories would strengthen the interpretative depth. The absence of emotional effects in older adults is consistent with positive effects, but alternative explanations deserve to be explored.
We thank the reviewer for this constructive comment. In the first section of the Discussion, we now more explicitly confront our findings with the predictions derived from the Cognitive Value Theory (CVT), clarifying how our results align with or diverge from this framework (p.15, 535-548).
Second, regarding age-related differences, we chose to focus on the most plausible and well-documented interpretations. Our discussion already considered several theoretically grounded accounts, namely age-related positivity effects, improved emotional regulation in older adults, and greater arithmetic expertise and automatization, which we believe provide the most parsimonious and empirically supported interpretations of our findings.
Conclusion and Implications: The conclusion could be strengthened by a clearer synthesis of the practical and theoretical implications. For example, the manuscript could address how distinct emotions differentially affect the strategic components of cognition, or how age-related resilience to negative emotional interference can underpin models of healthy cognitive aging and future research on strategic behavior.
We have revised the conclusion to place greater emphasis on both the empirical and theoretical implications of the study (p. 17-18).